The influence of a manipulation of threat on experimentally-induced secondary hyperalgesia

Bedwell Gillian J. 1 2
Louw Caron 2
http://orcid.org/0000-0003-4823-2487 Parker Romy 2
van den Broeke Emanuel 3
http://orcid.org/0000-0003-0437-6665 Vlaeyen Johan W. 4 5
http://orcid.org/0000-0002-3750-4945 Moseley G. Lorimer 6
http://orcid.org/0000-0002-5357-4062 Madden Victoria J. 2 4 6 7 torymadden@gmail.com
1 Department of Health and Rehabilitation Sciences, University of Cape Town , Cape Town, Western Cape , South Africa
2 Pain Unit, Department of Anaesthesia and Perioperative Medicine, Neuroscience Institute, University of Cape Town , Cape Town, Western Cape , South Africa
3 Institute of Neuroscience, Division Cognitive and Systems, UC Louvain , Brussels , Belgium
4 Research Group Health Psychology, KU Leuven , Leuven , Belgium
5 Experimental Health Psychology, University of Maastricht , Maastricht , Netherlands
6 IIMPACT in Health, University of South Australia , Adelaide , Australia
7 Department of Psychiatry and Mental Health, Neuroscience Institute, University of Cape Town , Cape Town, Western Cape , South Africa
La Touche Roy
Electronic publication date: 2022 Jun 20
Publication date: 2022
Volume: 10
Electronic Location ID: e13512
Received 2021 Oct 14; Accepted 2022 May 7
Copyright: © 2022 Bedwell et al.
Copyright year: 2022
Copyright holder: Bedwell et al.
License: This is an open access article distributed under the terms of the Creative Commons Attribution License, which permits unrestricted use, distribution, reproduction and adaptation in any medium and for any purpose provided that it is properly attributed. For attribution, the original author(s), title, publication source (PeerJ) and either DOI or URL of the article must be cited.
License URL: https://creativecommons.org/licenses/by/4.0/

Keywords: Pain, Threat, Secondary hyperalgesia, Healthy volunteers, Electrical stimulation, Mechanical hyperalgesia

Funding: University of Cape Town PainSA South African Society of Physiotherapy Pfizer National Research Fund (South Africa) Oppenheimer Memorial Trust Fonds de Recherche Clinique (FRC) of UCLouvain Methusalem Funding METH/15/011 National Health & Medical Research Council of Australia 1178444 South African National Research Foundation Fogarty International Center of the National Institutes of Health K43TW011442 IASP Developing Countries Collaborative Research Grant Gillian J. Bedwell was supported by a scholarship and postgraduate funding from the University of Cape Town, and Postgraduate Research Grants from PainSA and the South African Society of Physiotherapy, and an unrestricted education grant from Pfizer (2018) with no direct relationship to the current work. Gillian J. Bedwell is supported by a postgraduate scholarship from the National Research Fund (South Africa) and the Oppenheimer Memorial Trust. Emanuel van den Broeke was supported by the Fonds de Recherche Clinique (FRC) of UCLouvain, Belgium. Johan W. Vlaeyen is supported by the Asthenes research program “From Acute Aversive Sensations to Chronic Bodily Symptoms,” a long-term structural Methusalem funding (METH/15/011) from the Flemish government, Belgium. G. Lorimer Moseley was supported by a Leadership Investigator grant from the National Health & Medical Research Council of Australia (ID 1178444). Victoria J. Madden was supported by an Innovation postdoctoral fellowship from the South African National Research Foundation, and is supported by the Fogarty International Center of the National Institutes of Health (award K43TW011442). Research collaboration supported by the IASP Developing Countries Collaborative Research Grant.

==============================
Pain is thought to be influenced by the threat value of the particular context in which it occurs. However, the mechanisms by which a threat achieves this influence on pain are unclear. Here, we explore how threat influences experimentally-induced secondary hyperalgesia, which is thought to be a manifestation of central sensitization. We developed an experimental study to investigate the effect of a manipulation of threat on experimentally-induced secondary hyperalgesia in 26 healthy human adults (16 identifying as female; 10 as male). We induced secondary hyperalgesia at both forearms using high-frequency electrical stimulation. Prior to the induction, we used a previously successful method to manipulate threat of tissue damage at one forearm (threat site). The effect of the threat manipulation was determined by comparing participant-rated anxiety, perceived threat, and pain during the experimental induction of secondary hyperalgesia, between the threat and control sites. We hypothesized that the threat site would show greater secondary hyperalgesia (primary outcome) and greater surface area (secondary outcome) of induced secondary hyperalgesia than the control site. Despite a thorough piloting procedure to test the threat manipulation, our data showed no main effect of site on pain, anxiety, or threat ratings during high-frequency electrical stimulation. In the light of no difference in threat between sites, the primary and secondary hypotheses cannot be tested. We discuss reasons why we were unable to replicate the efficacy of this established threat manipulation in our sample, including: (1) competition between threats, (2) generalization of learned threat value, (3) safety cues, (4) trust, and requirements for participant safety, (5) sampling bias, (6) sample-specific habituation to threat, and (7) implausibility of (sham) skin examination and report. Better strategies to manipulate threat are required for further research on the mechanisms by which threat influences pain.

Introduction

Threat is thought to be important for pain. A growing body of research suggests that threat influences pain (Arntz & Claassens, 2004; Crombez et al., 1998; Karos et al., 2018; Reicherts et al., 2016; Wiech et al., 2010). There are many clinical examples of patients experiencing pain that is disproportionate to tissue damage and even pain in the absence of tissue damage (Caneiro et al., 2021; Fisher, Hassan & O’Connor, 1995; Flor, 2002; Melzack & Wall, 1965). This dissonance between pain and tissue damage is often attributed to threat (Caneiro et al., 2017; Moseley, 2007; Tabor et al., 2015).

The influence of threat on pain has been demonstrated in experimental studies that are assumed to have manipulated the perceived threat value of stimuli. The manipulations used include instructions about tissue vulnerability (Wiech et al., 2010), verbal instruction about stimulus intensity (Arntz & Claassens, 2004), visual cues that imply different threat values (Moseley & Arntz, 2007), and classical conditioning (Ploghaus et al., 2001). Interpretation of two of these studies (Arntz & Claassens, 2004; Moseley & Arntz, 2007) is hindered by lack of evidence that threat value was actually manipulated. Selecting an appropriate test for a change in threat is difficult. Wiech et al. (2010) interpreted a difference in pain ratings as an indication of differential threat, revealing the assumption that pain and threat are linked. Ploghaus et al. (2001) assessed whether their threat manipulation changed self-reported anxiety ratings and heart rate, perhaps more arguably capturing change in affect and physiology that would be expected to change with threat. Despite the difficulties confirming manipulation efficacy, the idea that threat influences pain is broadly accepted.

The exact mechanisms by which threat influences pain are unclear. It is possible that the mechanism is located largely within the brain. Several imaging studies suggest that anxiety about a threatening stimulus is linked to greater pre-stimulus activity in key brain regions such as the anterior insula, midcingulate cortex, and hippocampus—and that this activity is associated with pain to that stimulus (Ploghaus et al., 2001; Ploner et al., 2010; Wiech et al., 2010). Computational modelling of cognitive decisions about pain have demonstrated that prior information about an event (e.g. state of the body vs. danger posed by a stimulus) can influence the painfulness of that event (Wiech et al., 2014; Zaman et al., 2017)—indeed, the anterior insula is thought to be closely involved in interoception and therefore informing priors about body-related events (Craig & Craig, 2009; Ploner et al., 2010).

Further, it is similarly possible that threat could influence pain not only via brain-dominant processes, but also by altering spinal processing of nociception. Descending modulation can influence synaptic transmission of nociception at the dorsal horn of the spinal cord (Gebhart, 2004; Porreca, Ossipov & Gebhart, 2002; Ren & Dubner, 2002; Suzuki, Rygh & Dickenson, 2004; Urban & Gebhart, 1999). Descending inhibition is enacted via descending monoaminergic pathways that use serotonin, noradrenaline, and dopamine (Gebhart, 2004; Millan, 2002; Pertovaara, 2006; Zhao et al., 2007). Contextual threat influences descending inhibition (Moseley & Arntz, 2007) and may account for the Beecher’s (1946) soldiers reporting diminished pain severity despite presenting with extensive tissue damage. Another example is the 37% of patients presenting to the emergency unit who report a pain-free period of 1 to 9 h after injury (Melzack, Wall & Ty, 1982). These examples are of pain diminution presumed to arise by descending inhibition—and, here, lack of pain is thought to support survival. In contrast, threat of tissue damage may promote descending facilitation to increase pain and motivate protection of the (potentially) damaged tissue. In this study, we aimed to establish whether threat of tissue damage increases spinal facilitation of nociception.

Experimentally induced secondary hyperalgesia provides a useful model of spinal facilitation of nociception. Secondary hyperalgesia is defined as “increased pain from a stimulus that normally provokes pain” outside the area of tissue damage (Merskey & Bogduk, 2017). Secondary hyperalgesia is thought to be mediated by an altered response profile of dorsal horn neurons. Experimental induction of secondary hyperalgesia uses safe stimulation to induce a short-lived expression of secondary hyperalgesia under controlled conditions, in a laboratory. The induction can use stimuli such as high-frequency electrical stimulation (HFS) (Klein et al., 2004), low-frequency electrical stimulation (Torta et al., 2019), intradermal capsaicin injection (Baron et al., 1999), topical capsaicin application (You, Creech & Meagher, 2016) and burn injury (Wahl et al., 2019). In this study, we used high-frequency electrical stimulation to induce experimental secondary hyperalgesia.

In the current study, we aimed to manipulate threat of tissue damage using a (sham) skin examination and report, and to test the influence of that manipulation of threat on the magnitude (primary outcome) and surface area (secondary outcome) of experimentally induced secondary hyperalgesia. We hypothesized that the magnitude (primary outcome) and surface area (secondary outcome) of induced secondary hyperalgesia would be greater at the threat site than at the control site.

Methods

Study design

The protocol and the pilot analysis were preregistered with Open Science Framework at https://osf.io/nk2hj/ to ensure detailed documentation of the research process, thus supporting accountability and study replication (Lee et al., 2018; Lindsay, Simons & Lilienfeld, 2016). The study was designed as a within-subject, double-blinded experiment. It was conducted at the University of Cape Town, South Africa. The protocol was approved by the Faculty of Health Sciences Human Research Ethics Committee (REF 498/2018), University of Cape Town. An extensive piloting procedure was conducted to ensure the effectiveness of the threat manipulation procedure (File S1). Data were collected between October and November 2019.

Participants

Volunteers were recruited from the general public using advertisements, social media channels such as Facebook, and word of mouth. Volunteers were sent the study information sheet describing the details of the procedure via email, and were screened for exclusion criteria by completing an online eligibility quiz using the Responster platform (https://responster.com). After completing the screening quiz, eligible volunteers were contacted via email to organize a booking. Participants were able to withdraw from the study at any stage during the procedure or up to 48 h after the procedure. They were compensated ZAR1001 , in cash, for their time and inconvenience, even if they withdrew from the study.

Inclusion and exclusion criteria

Volunteers needed to be healthy, pain-free adults between the ages of 18–65, able to provide written consent autonomously, and fluent in speaking, understanding, and reading English (all as per volunteers’ self-reports). Volunteers were excluded from the study if they reported one or more of the following: chronic pain—pain for most days of the week for the past 3 months (Blyth et al., 2001), pain on the day of testing, self-reported pregnancy, electronic implant (e.g. pacemaker), any kind of heart/cardiovascular problem, diabetes mellitus, neurological problems (e.g. epilepsy), peripheral vascular disease, problems with skin healing, use of analgesics within 24 h before testing, use of medication that could alter skin sensitivity or healing (e.g. analgesic medication, topical medical creams or immune modulators), history of psychiatric problems (e.g. fear or anxiety disorder, or clinical depression), and previous participation in this study or a closely related study. Additionally, volunteers with upper limb tattoos distal to the anode were ineligible to participate as some tattoo inks contain metals and therefore pose a small risk of electrical conductance (Ross & Matava, 2011) and skin burn.

Randomization and blinding

This study was designed for blinding of assessor, participants, and data analyst.

Blinding of assessor

VJM conducted concealed allocation of arm to condition, i.e. which arm (left or right) would receive the HFS under a condition of threat. The allocation of arm to condition was counterbalanced, as follows. First, 13 rows of each of Group 1 (threat site: right arm) and Group 2 (threat site: left arm) were entered into Excel to account for the total planned sample size of 26 (see Sample size calculations below). A random number was generated for each row. The list was then re-ordered using the random numbers and this new list order was locked. Second, papers stating either ‘Group 1’ or ‘Group 2’ (13 for each group) were placed into 26 sequentially numbered, opaque envelopes, in accordance with the locked list order. Third, the envelopes were used in the order specified by the numbers written on them.

CL (the assessor) conducted the experimental procedure and sensory testing for all participants. She gave the sealed envelope to the research assistant, who opened the envelope and allocated the condition in the software program while the assessor was outside the room. The assessor was thus unaware of participants’ condition (i.e. which arm right/left would receive the HFS under a condition of threat). This mitigated verification bias. Given that the assessor was aware of the aims of the study, they completed a blinding assessment after participants received the HFS and before the sensory testing battery. The blinding assessment required the assessor to state (or guess) which arm had received the HFS under a condition of threat, and to rate her confidence about this on a Likert scale (“not at all confident”, “not confident”, “neutral”, “confident”, “extremely confident”).

Blinding of participants

Participants were informed that the study investigated how people experience painful and non-painful stimulations. No details of the aims or hypotheses were provided, to maintain participant blinding.

Blinding of data analyst

GJB performed the statistical analyses and was blinded to condition while conducting the analyses. VJM re-coded participants’ condition allocation prior to statistical analysis to ensure blinding of GJB to condition. The allocation of arm to condition was re-coded to “Condition A” or “Condition B”, such that condition A denoted the condition of threat and condition B denoted the safe condition. Conditions A and B were interpreted by GJB after all analyses were completed.

Equipment

HFS was provided using a constant current stimulator (DS7A; Digitimer Limited, Hertfordshire, UK) controlled by Affect5 software (Spruyt et al., 2010). Current was directed from the DS7A, via a D188 (Digitimer Limited, Hertfordshire, UK), to two pairs of electrodes. The electrodes consisted of two cathodes and two anodes. The cathodes had 10 blunt steel pins arranged in a circle and were secured to both anterior forearms. The anodes were large, flexible surface electrodes and were secured to both upper arms (File S2). The cathodes were secured on the anterior aspects of both the participant’s forearms, with a double-sided sticker, approximately eight centimeters distal to the cubital fossa, and avoiding any visibly prominent vasculature. Large surface electrodes were placed around both upper arms and served as the anodes.

Manipulated variables

High-frequency electrical stimulation

Participants received HFS on both forearms, asynchronously. HFS was delivered to one arm under a condition of threat (threat site) and to the other arm under a neutral condition (control site), thus providing a within-subject comparison.

The appropriate intensity of the HFS depends on the electrode used and individuals’ electrical detection threshold. The electrodes in this current study most closely resembled those used by Klein et al. (2004), Klein et al. (2008) and Henrich et al. (2015). Their work and our pilot have shown induction of robust secondary hyperalgesia with HFS delivered at 100 Hz, at a current of 10 times the individual detection threshold.

Participants were orientated to the electrical stimulus (refer to the Procedure section) and the stimulus was calibrated to the participant’s individual electrical detection threshold. This calibration consisted of single electrical stimuli, with a pulse width of 2 ms. An adaptive staircase approach (see Procedure below) was used to determine the individual electrical detection threshold. The electrical detection threshold was used to determine the current of the HFS at 10 times the electrical detection threshold. Klein et al. (2004) reported participants’ electrical detection threshold to be 0.11 ± 0.06 mA (mean ± SD). Therefore, it was anticipated that the range of currents would be similar in this current study.

The HFS consisted of five 1-s trains, using 2 ms pulse width, of 100 Hz frequency, with a 9-s break between trains (Klein et al., 2004; Pfau et al., 2011; van den Broeke & Mouraux, 2014). The current of the stimulation was 10 times the participant’s individual electrical detection threshold.

Threat manipulation

The threat manipulation procedure was modelled on that used by Wiech et al. (2010) and consisted of a sham skin examination and report. Our sham skin examination and report were conducted after the baseline sensory assessment and before participants received the HFS. The assessor informed participants that she was examining the robustness of the skin around the electrodes, to determine the risk of skin damage associated with HFS. She used an otoscope to magnify and illuminate the skin. She then left the room to ostensibly enter the (sham) examination results into the computer for it to apply an “algorithm” to determine the skin’s safety. Finally, the sham results were shown to the participant on a screen not visible to the assessor. For each participant, the threat site was deemed “approved with reservations” on the screen, with participants instructed to monitor their “fragile” skin closely during the HFS as there was “moderate risk of injury”. For the control site, “fully approved” was reported on the screen, with participants informed that the skin is “robust” and there was “low risk of injury” during the HFS.

Threat manipulation check

Three manipulation checks were performed to determine the effectiveness of the sham skin examination and report: (1) five Sensation and Pain Rating Scale (SPARS) ratings during the HFS induction (one for each train) were compared between the threat and control sites, and a customized questionnaire was used to assess (2) self-reported anxiety and (3) self-reported threat of tissue damage during the HFS induction. We opted to include these three manipulation checks to provide insight into both an expected effect of implicit threat (pain ratings during HFS) and explicit threat (self-reported anxiety and threat of tissue damage). After locking the protocol online, we realized we had not designated any of the three participant-reported manipulation check outcomes as primary. Given that our manipulation was based on that of Wiech et al. (2010), who used reported experimental pain as the manipulation check, we designated SPARS ratings during the induction as the primary outcome for the manipulation check in the current study.

Self-reported anxiety and self-reported threat of tissue damage were assessed at the end of the experiment. Participants were asked to indicate on a five-point Likert scale the extent to which they agreed or disagreed with the following statements: “At the time of receiving the intense electrical stimulation on my right/left arm, I felt anxious” (i.e. self-reported anxiety) and “At the time of receiving the intense electrical stimulation, I was concerned that it would cause damage to my skin on my right/left arm” (i.e. self-reported threat of tissue damage). Participants completed these questions with reference to each arm separately.

Measured variables

Participants verbally reported sensation or pain ratings using the SPARS (File S3) (Madden et al., 2019). This scale provides for a range of non-painful and painful sensory experiences. The non-painful range, on the left-hand side of the scale, ranges from −50—“no sensation”— to 0—“the exact point at which what you feel transitions to pain”. The painful range, on the right-hand side of the scale, ranges from 0 to +50—“most intense pain you can imagine”. The SPARS is sensitive to change in both painful and non-painful sensory experiences (Madden et al., 2019).

Outcomes

Primary outcome

Mechanical punctate stimulation

Mechanical punctate stimulation was provided with two pinprick stimulators (MRS Systems, Heidelberg, Germany), exerting forces of 128 and 256 mN, respectively. Participants were asked to close their eyes while the assessor provided three stimulations at 1 s intervals within a one-centimeter radius around the electrode with each pinprick stimulator. Participants were asked to provide an average SPARS rating for the three stimulations for each pinprick stimulator separately (i.e. an average rating for the three stimulations from the 128 mN and an average ratings for the three stimulations from the 256 mN pinprick stimulator). Increased SPARS ratings to these modalities in the region surrounding the distal electrode, after the HFS, indicated the presence of secondary hyperalgesia. We were not interested in the effect of force as a predictor in this current study. Therefore, we used the mean SPARS ratings of the two different pinprick weights to determine the overall mechanical punctate stimulation rating, instead of including force as a predictor variable. This kept the model simple to maximize power.

Secondary outcome

Mapping surface area of secondary hyperalgesia

The area of secondary hyperalgesia was mapped using the eight-radial-lines approach, where eight lines are mapped at 45° angles (File S4) using a pinprick stimulator exerting a force of 128 mN (You et al., 2014). First, the assessor screened for the presence of secondary hyperalgesia. This was performed by asking participants if they felt “a very obvious difference in sensation” when applying the pinprick stimulator at the most distal dot compared to the most proximal dot on the proximal-distal radial line (E in File S4). This process was repeated by stimulating the most proximal dot first and most distal dot on the same line. If participants still reported no difference in sensation, we concluded that the surface area for secondary hyperalgesia was zero at that time point; if participants reported a difference, then the assessor mapped the surface area. Briefly, the assessor provided a single stimulation at each point on a radial line, moving from the most distal point from the electrode towards the electrode. The participant was asked to report the point at which they felt a distinct change in sensation from the current stimulation, compared to the previous stimulation. This is interpreted as indicating the boundary of secondary hyperalgesia. This procedure was repeated along each of the eight radial lines, to obtain eight points of transition. The surface area thus identified comprises eight 45° triangles. We calculated and summed the surface area of the eight triangles using the equation surface area ½ab (sin 45°) (where a and b are the lengths of the sides of a triangle adjacent to the 45° angle), to calculate the overall surface area of secondary hyperalgesia.

Exploratory outcomes

Data obtained from assessing static light touch, dynamic light touch and single electrical stimulation were used for exploratory purposes only. Static light touch sensation was assessed with application of a von Frey filament (MARSTOCK, Schriesheim, Germany) that exerted a force of 32 mN upon bending (Rolke et al., 2006). Dynamic light touch was measured using a cotton wisp and soft brush stroke (Henrich et al., 2015). The electrical stimulus was 2 ms long with an intensity of 10 times the individual’s electrical detection threshold (Henrich et al., 2015). The results of these exploratory outcomes are provided in File S5.

Questionnaires

A history of previous trauma has been associated with increased area of secondary hyperalgesia (You, Creech & Meagher, 2016): women reporting childhood trauma and/or recent trauma displayed a larger surface area of secondary hyperalgesia after application of topical capsaicin than women reporting no history of trauma (You, Creech & Meagher, 2016). Therefore, we assessed childhood trauma and adult trauma using the Childhood Trauma Questionnaire (CTQ) (Bernstein et al., 2003) and a modified version of the World Mental Health Survey Initiative version of the World Health Organization’s Composite International Diagnostic Interview for post-traumatic stress disorder (WMH-CIDI) (Kessler & Üstün, 2004). The CTQ focuses on five criteria: emotional abuse, physical abuse, sexual abuse, emotional neglect, and physical neglect. The modified version of the WMH-CIDI screens for specific traumatic events. The WMH-CIDI was used as an inventory, therefore we did not investigate the details of the traumatic event(s). These data were used in a secondary analysis, to investigate the relationship between a history of trauma and experimentally induced secondary hyperalgesia, in an attempt to replicate the work by You, Creech & Meagher (2016).

Participants also completed several other questionnaires: 10-item Connor–Davidson Resilience Scale (Connor & Davidson, 2003), Cohen’s Perceived Stress Scale, Pain Catastrophizing Scale (Sullivan, Bishop & Pivik, 1995), Multidimensional Scale of Perceived Social Support (López-Martínez, Esteve-Zarazaga & Ramírez-Maestre, 2008), and 16-item Pain Vigilance and Awareness Questionnaire (McCracken, 1997), which were used for exploratory analyses to inform the development of future research questions only. The results for these questionnaires will not be reported here.

Procedure

Overview of procedure

The procedure was conducted in a quiet room in the Department of Anaesthesia and Perioperative Medicine, Groote Schuur Hospital, Cape Town. The procedure lasted approximately 2 h. An overview of the procedure is described in Fig. 1. First, participants underwent three rounds of baseline sensory testing. Second, the assessor performed that (sham) skin examination and report (i.e. threat manipulation). Third, participants received the HFS for ~1 min on each arm separately. Fourth, participants completed questionnaires during a 20-min break. Finally, 20 min after the HFS induction, the assessor performed repeated sensory follow up testing in 6-min intervals and surface area mapping in 20-min intervals. Last, participants were debriefed on the threat manipulation and reassured about the safety of the procedure.

Figure 1 Study procedure.

Preparation

The assessor used a formal script (File S6) during the procedure to standardize all the information presented to participants. When each participant arrived for testing, they were asked to re-read the study information sheet, confirm that none of the exclusion criteria applied, and sign the document of informed consent. Thereafter, participants were asked to remove any jewelry from their arms and to turn off mobile devices. The assessor used a stencil to mark locations for the electrodes, and to mark the eight radial lines on the participant’s skin. The assessor secured the electrodes in place using a double-sided electrode sticker.

Participants were orientated to the SPARS and the sensory testing battery. This orientation consisted of an explanation of how to use the SPARS and a brief demonstration of each of the sensory tests on the assessor’s arm. Participants had an opportunity to practice using the SPARS while the assessor ran through a practice round of the sensory testing battery.

Individual electrical detection threshold

Participants were orientated to the electrical stimulus and the stimulus was calibrated to their individual electrical detection threshold on both arms. The intensity started at zero and slowly increased in 0.1 mA increments until the participants reported that they could feel the electrical stimulus. They were informed that the electrical stimulus would “feel like a very tiny pinprick”. Participants were asked to say “yes” if they felt it, even slightly. This adaptive staircase approach was used to determine the individual electrical detection threshold on both arms. We used the average of the individual electrical detection thresholds from both arms for the HFS procedure.

Baseline testing

Once the participants were comfortable with using the SPARS, the sensory testing battery was conducted three times on each arm to obtain a stable estimate of baseline sensory ratings. Initially, the protocol outlined that primary hyperalgesia would not be assessed at this time point, as the electrical stimulation would not yet be calibrated to the participant. This was an error in the protocol and baseline primary hyperalgesia was assessed (protocol deviation one of three). The area of secondary hyperalgesia was not mapped at this point as secondary hyperalgesia had not yet been induced by HFS.

Sham skin examination

Next, the sham skin examination was performed, and the report provided.

High-frequency electrical stimulation

Before the HFS was delivered, participants were thoroughly briefed on what to expect from the HFS. Participants were informed that most people find the HFS “moderately painful” and they may withdraw with immediate effect at any point during the procedure. They were instructed to say “STOP” if they wished to withdraw, in which case the assessor would use the safety switch on the stimulator to deactivate the stimulator immediately. Participants were asked to provide ratings using the SPARS after each HFS train. As mentioned, the HFS SPARS ratings served as one of three threat manipulation checks.

Waiting period

There was a waiting period of 20 min after the HFS to allow time for the secondary hyperalgesia to develop. To optimize time, this period was used to administer the seven questionnaires (not reported on here).

Follow-up testing

The battery of sensory testing was conducted every 6 min from 20 to 56 min after the HFS, to capture the development of secondary hyperalgesia, the timing of which can vary between individuals (Pfau et al., 2011). The order of the sensory testing modalities was randomized within each time point, for each participant, to decrease predictability and ensure accurate ratings (with the same order used for both arms, within each time point). The surface area of secondary hyperalgesia was mapped at 20, 40, and 60 min after the HFS.

Post-experiment questionnaire and debriefing

After the follow-up testing, the electrodes were removed, and participants completed the post-experiment questionnaire assessing self-reported anxiety and self-reported threat of tissue damage. These two questionnaires served as a second and third threat manipulation check. After the threat manipulation check, the assessor also conducted a semi-structured interview where participants were asked to elaborate on their answers for their self-reported anxiety and threat of tissue damage during HFS induction. The assessor wrote down direct quotes of participants’ responses. This semi-structured interview was planned after the protocol had been locked online and therefore was not included in the protocol (protocol deviation two of three). These responses were used to gain further insight into the effectiveness of the threat manipulation. Finally, participants were debriefed on the threat manipulation and reassured about the safety of the procedure.

Participants completed all the questionnaires in privacy, and on a computer. Details of any traumatic events was not requested. For these reasons, together with the strict eligibility criteria, it was unlikely the questionnaires would have evoked strong emotional responses at the time of testing. Nevertheless, after the procedure, participants were provided with an information pamphlet listing the local non-profit organizations where they could access psychological assistance, if they wished to do so. Additionally, all participants received a list of the community health care centers in Cape Town that provide psychological counselling as well as a list of two private practice psychologists within the University of Cape Town’s neighboring communities.

Statistical analysis

Sample size calculation

Pilot data and the GLIMMPSE online calculator (Kreidler et al., 2013) were used to estimate the sample size required to achieve 80% power to detect a minimum 5-unit difference (on a 100-unit scale) in secondary hyperalgesia, with alpha set at 0.05. A mixed linear regression was planned, in which the dependent variable was the mean rating to both pinprick stimulators (128 and 256 mN) at each time point after HFS, minus the equivalent mean rating at the baseline time point (before HFS). The model structure allowed each participant to have their own intercept (i.e. individual participant (ID) was a random factor). The independent variable, ‘condition’ (i.e. threat or control site), was a fixed factor, and the repeated measures variable ‘time’ was nested within and fully crossed with participant ID, because each participant was assessed at each time point. In the lme4 package (Bates et al., 2015; Loy & Hofmann, 2014) of R (version 3.5.3 (2019-03-11)), the model structure was: lmer(rating ~ condition + (1|ID/time)). The scale factor in GLIMMPSE influenced the estimated sample size required; therefore, we used the pilot data (n = 6, from the third phase of piloting) to guide our choice of scale factor. In fact, our pilot data were best described by a beta:sigma relationship of 1:2; however, given the small sample size for our pilot data and to minimize the risk of missing an effect that might exist, we opted to use the sample size recommended for a beta:sigma relationship of 0.5:1.5. GLIMMPSE estimated that a total sample size of 25 participants was required to detect a main effect of condition. Therefore, a sample size of 26 was used to allow for counterbalancing for the manipulation site.

Preliminary assessment of the data

It was plausible that the individual calibration approach could have confounded the results because the current for the HFS was linked to the individual detection threshold, and HFS delivered at a higher current could result in greater secondary hyperalgesia. Although previous published datasets (Torta et al., 2017; van den Broeke et al., 2017; van den Broeke et al., 2019; van den Broeke et al., 2019; Van den Broeke et al., 2016; van den Broeke, Lenoir & Mouraux, 2016) (n = 170, unpublished) investigation found no association between the individual electrical detection threshold and the magnitude of secondary hyperalgesia. We checked this by testing for a correlation between the individually determined electrical detection threshold and magnitude of secondary hyperalgesia in our data.

Analysis of blinding assessment for the assessor

An analysis strategy for assessing assessor blinding was not specified in the protocol. Post hoc, we opted to calculate the percentage of correct guesses of site allocation by the assessor. If the percentage correct was greater than 50%, we planned to use the data from the confidence scale to explore a percentage greater than 50% (the defined limit) in terms of confidence, to work out the likelihood that the percentage was due to genuine guessing.

Analysis of the manipulation checks

The effect of the manipulation was assessed by comparing (1) pain ratings during the HFS induction (primary indicator), (2) self-reported anxiety, and (3) self-reported fear of tissue damage for each arm owing to the HFS induction. A mixed model analysis was used to compare ratings of the HFS trains (rating ~ condition + (1|id)), anxiety (anxiety ~ condition + (1|id)) and threat of tissue damage (threat ~ condition + (1|id)) between conditions. A main effect of condition on ratings, anxiety and threat of tissue damage would confirm the efficacy of the manipulation. The models allowed for each participant to have a different intercept.

Primary analysis

Response data were analyzed using linear mixed modelling, to account for individual variability in responses whilst still testing for a between-site effect at the group level. The study was designed to have within-subject controls of both pre- and post-induction measurements and control site measurements. Therefore, the change in sensitivity (pre-induction measurements subtracted from post-induction measurements) was compared between arms (within subjects). The exact parameters for the analysis were chosen based on visual inspection of the data (including an assessment of distribution), and the appropriate tests to confirm or refute any assumptions of the analytical strategy. As specified above, the primary outcome was the magnitude of secondary hyperalgesia.

The planned data analysis was finalized using the full pilot study data. The sensory ratings and questionnaire data had been imported into R data frames prior to the protocol being locked online, but no exploratory plotting or analyses had been done at this stage. Data analysis commenced only after the protocol had been locked online. The pilot analysis was not substantively changed after initial processing of the formal data commenced, except that the assessment of model fit was added, having been omitted from the pilot data analysis (protocol deviation of three of three).

A robust mixed linear modelling approach, using the ‘lmer’ option Satterthwaite approximation within the lmer package (Kuznetsova, Brockhoff & Christensen, 2017), was used for the formal data analysis. This allowed for both random effects (participant) and fixed effects (site), as used in our sample size calculation. Two models were tested for this analysis: the first was a fully crossed model with ID (rating_controlled ~condition + (1|id/time)); the second was one in which that assumption was not made (rating_controlled ~ condition + (1|id)). ‘Fully crossed with’ means that every time point was assessed for every ID. This was indeed the case in this present study’s design. Therefore, the fully crossed model most closely represents the design of this experiment. The fully crossed model was compared to the null version of the model (rating_controlled ~ (1|id)) (which does not include condition as a predictor variable). If the ANOVA that compared two models (e.g. fully crossed and null version) yielded a significant p-value, the interpretation was that the non-null (e.g. fully crossed) model fits the data better than the null.

Secondary analysis

A secondary analysis investigated the relationship between a history of trauma and the surface area showing experimentally induced secondary hyperalgesia, replicating the work by You, Creech & Meagher (2016). In their study, they summed the results of participants’ individual scores from the CTQ and the Recent Traumatic Events Scale to obtain an individual stressful life events score. Similarly, in this current study the results of the CTQ and WMH-CIDI were summed. You, Creech & Meagher (2016) reported a larger surface area and magnitude of capsaicin-induced secondary hyperalgesia in participants with a history of trauma than participants without a history of trauma. In this current study, a univariate regression was conducted to examine whether stressful life events correlate with the area of secondary hyperalgesia in this sample.

Assessment of model fit

An assessment of model fit was conducted for both the primary and secondary analyses. Two assumptions were assessed. If either of the assumptions was violated, the model was deemed unfit. The two assumptions were: (1) homoscedasticity and (2) normally distributed residuals (Winter, 2013).

Results

Data were analyzed using R (version 4.1.1, packages: tidyverse (Wickham et al., 2019), magrittr (Milton Bache & Wickham, 2014), ggplot2 (Wickham, 2016), readxl (Wickham & Bryan, 2019), lme4 (Bates et al., 2015), gridExtra (Auguie, 2017), grid (R Core Team, 2020), lmerTest (Kuznetsova, Brockhoff & Christensen, 2017), and here (Müller, 2017)) in RStudio (RStudio Team, 2019). Results are presented using box-and-scatter plots created with ggplot2. Boxplot whiskers represent the maximum and minimum values, the ends of the box represent the upper and lower quartiles, and the horizontal line within the box represents the median.

Participants

Forty people volunteered to participate in this study and completed the eligibility quiz. Fourteen volunteers were excluded for: tattoos distal to the anode (n = 5), chronic pain (n = 5), history of mental illness (n = 3), and being unavailable for testing (n = 1). A sample of 26 (16 females) was used for this study. The median age was 21 (range 18–55) years old. No participants withdrew from the study. None of the participants reported taking analgesic medication prior to the procedure. We did not assess for adverse events; however, no adverse events were reported by the assessor and participants.

Assessing for confounding of the magnitude of secondary hyperalgesia by current

The mean (±SD) individual electrical detection threshold for the HFS procedure was 1.60 mA (±0.64 mA). The Shapiro–Wilk test showed that the data on peak secondary hyperalgesia magnitude were not normally distributed (p = 0.011). Therefore, a Spearman rank-order correlation test was used to check for a relationship between the calibration current and the peak magnitude of secondary hyperalgesia. There was no significant correlation between the calibration current and the peak magnitude of secondary hyperalgesia (rho = 0.040; p = 0.78).

Blinding assessment of the researchers conducting the experiment

The assessor correctly guessed site allocation 42.31% of the time. A plot showing the relationship between confidence level and accuracy of guessing condition allocation can be seen in File S5.

Manipulation checks

HFS intensity ratings

All HFS trains were rated in the painful range of the SPARS: mean ± SD 38.77 (±11.34) at the threat site and 39.07 (±11.31) at the control site (Fig. 2). There was no main effect of condition on SPARS ratings of the HFS trains (p = 0.646).

Figure 2 Individual SPARS ratings of each HFS train (5 trains × 26 participants) delivered to the control site (green) and the threat site (red).

Each dot represents a rating from one participant for one train. The SPARS has a non-painful range between −50 and 0, however, only the painful range (+5 to +50) is shown here because all train ratings were in the ‘painful’ range. Boxplot whiskers represent the maximum and minimum values, the ends of the box represent the upper and lower quartiles, and the horizontal line within the box represents the median.

Self-reported anxiety during HFS

The mean (±SD) anxiety ratings were 3.31 (±1.12) for the threat site and 3.42 (±1.13) for the control site, out of a maximum of five. There was no main effect of condition on anxiety ratings (p = 0.31) (Fig. 3).

Figure 3 The relationship between condition and anxiety rating (n = 26).

Anxiety rating reflects response to the statement, “At the time of receiving the intense electrical stimulation on my right/left arm, I felt anxious”, where 1 = strongly disagree and 5 = strongly agree. Each dot represents each participant’s response with reference to the control site (green) and the threat site (red), with horizontal jitter added to aid visibility. Boxplot whiskers represent the maximum and minimum values, the ends of the box represent the upper and lower quartiles, and the horizontal line within the box represents the median.

Self-reported threat of tissue damage during HFS

The mean (±SD) threat of tissue damage ratings was 2.81 (±1.30) for the threat site and 2.50 (±1.14) for the control site. There was no main effect of condition on threat ratings (p = 0.11) (Fig. 4).

Figure 4 The relationship between condition and threat rating n = 26.

Threat rating reflects responses to the statement “At the time of receiving the intense electrical stimulation, I was concerned that it would cause damage to my skin on my right/left arm”, where 1 = strongly disagree and 5 = strongly agree. Each dot represents each participant’s response with reference to the control site (green) and the threat site (red). Boxplot whiskers represent the maximum and minimum values, the ends of the box represent the upper and lower quartiles, and the horizontal line within the box represents the median.

Primary analysis

Primary outcome: mechanical punctate stimulation

The primary aim of this study was to test the influence of a manipulation of threat on magnitude of secondary hyperalgesia. Figure 5 displays the magnitude of secondary hyperalgesia over time, grouped by condition. There was no main effect of condition on the magnitude of secondary hyperalgesia (p = 0.73) (Fig. 6).

Figure 5 Magnitude of secondary hyperalgesia at each time point, by condition.

Each dot represents the SPARS rating to pinprick at the control site (green) and threat site (red) at each time point for each participant, with the exception that each dot at time −4 represents a mean of three baseline trials. A negative SPARS rating indicates that the pinprick stimulus was non-painful; a positive SPARS rating indicates that the pinprick stimulus was painful. The vertical orange line shows the time of induction, which was 20 min before the first follow-up time point. The horizontal blue line represents ratings of 0—the exact point of transition from non-painful to painful. Boxplot whiskers represent the maximum and minimum values, the ends of the box represent the upper and lower quartiles, and the horizontal line within the box represents the median.

Figure 6 Between-condition difference in magnitude of secondary hyperalgesia at each time point, within each participant (n = 26).

Each dot represents the difference for one participant at each time point. The vertical orange line represents the time of HFS induction, which was 20 min before the first follow-up time point. The horizontal blue line represents ratings of 0—the exact point at which ratings transition from non-painful to painful. Boxplot whiskers represent the maximum and minimum values, the ends of the box represent the upper and lower quartiles, and the horizontal line within the box represents the median.

Assessment of model fit

The best model had the structure: difference_in_ratings ~ condition + (1|id/time). First, the assumption of homoscedasticity was assessed, i.e. assessment for equal variance across the range of predicted values. There was slightly increased density on the left, but the range of the maximum and minimum values seemed consistent across the x-axis. Therefore, the assumption of homoscedasticity was deemed to have been upheld. Second, the assumption that residuals were normally distributed was assessed. The Q-Q plot showed minor deviations from the diagonal reference line and the histogram showed acceptable distribution. Therefore, the assumption that residuals were normally distributed was deemed to have been upheld. In conclusion, both assumptions were upheld by the data, suggesting that the model could be used.

Secondary outcome

Surface area of secondary hyperalgesia

The secondary aim was to test the influence of a manipulation of threat on surface area of secondary hyperalgesia. Figure 7 displays the mean area of secondary hyperalgesia for each time point. Secondary hyperalgesia surface area was not predicted by condition (p = 0.16).

Figure 7 Surface area of secondary hyperalgesia for each time point, by condition, and within participant (n = 26).

Each dot represents the surface area at the control site (green) or threat site (red) at each time point for each participant. Boxplot whiskers represent the maximum and minimum values, the ends of the box represent the upper and lower quartiles, and the horizontal line within the box represents the median.

Assessment of model fit

The best model had the structure: surface_area ~ condition + (1|id/time). First, assumption of homoscedasticity was assessed, i.e. assessment for equal variance across the range of predicted values. Slightly increased density in in the middle and slightly smaller ranges of the maximum and minimum values on the left than on the right were considered inconsequential given the robust nature of the lmer method (Loy & Hofmann, 2014). Therefore, the assumption of homoscedasticity was deemed to have been upheld. Second, the assumption that residuals were normally distributed was assessed. The Q-Q plot and the histogram shows normal distribution. Therefore, the assumption that residuals were normally distributed was deemed to have been upheld. In conclusion, both assumptions were upheld by the data, suggesting that the model could be used.

Semi-structured interview

In general, participants reported being more anxious about the pain associated with the HFS induction than about the results of the (sham) skin examination. Seven (of 26) participants also reported trusting that enough precautions had been taken to ensure the safety of the procedure (File S6).

Planned exploratory analysis: the relationship between trauma scores and surface area of secondary hyperalgesia

A Shapiro–Wilk test showed that the data were normally distributed (p = 0.48); therefore, a Pearson’s correlation test was used. There was no statistically significant correlation between summed trauma score and surface area of secondary hyperalgesia (p = 0.16) (File S5).

Discussion

This study aimed to investigate the influence of a manipulation of threat on magnitude (primary outcome) and surface area (secondary outcome) of experimentally-induced secondary hyperalgesia in healthy human volunteers. We hypothesized that the threat site would show greater secondary hyperalgesia (primary outcome) and greater surface area (secondary outcome) of induced secondary hyperalgesia than the control site. Despite careful development and pilot-testing of the threat manipulation, it showed no differential effect in this study. Given no difference in threat between sites, it is unsurprising that the primary analysis did not show a main effect of condition on magnitude and surface area of secondary hyperalgesia.

Threat manipulation

It is surprising that the current threat manipulation was ineffective, given that it was based on a manipulation previously thought to be effective as a threat manipulation (Wiech et al., 2010). We identified two possible explanations. On the one hand, the threat may have been manipulated, but to the same extent in both conditions—which would have been missed by our between-condition manipulation check. On the other hand, the threat may truly have been unmanipulated. We discuss both possibilities here. First, we consider two processes by which threat could have been altered to the same extent in both conditions: (1) competition between threats and (2) generalization of learned threat value.

Competition between threats

Anticipated painfulness of the HFS (which was applied to both arms) may have competed with and exceeded the threat of tissue damage (which was applied to only one arm—threat site). Eleven (of 26) participants reported feeling more anxious about anticipating the pain associated with the HFS than about possible tissue damage—and, indeed, the painfulness of the HFS may have been a more immediate threat than tissue damage. Our analyses were designed to detect a difference between arms, so neither our manipulation checks nor our primary analysis would have detected possible bilateral modulation of secondary hyperalgesia by threat. However, an unplanned exploratory analysis indicated a positive correlation between threat ratings at the two sites, which provides preliminary support for this possibility (File S5).

Generalization of learned threat value

It is also possible that participants generalized the learned threat value of the first induction to the second induction, regardless of condition. However, exploratory analysis of the manipulation check data (File S5) revealed no evidence of an order effect on ratings of HFS intensity, anxiety, or threat of tissue damage.

Next, we consider five influences that could have prevented any manipulation of threat: (1) safety cues, (2) trust, and requirements for participant safety, (3) sampling bias, (4) sample-specific habituation to threat and (5) implausible (sham) skin examination and report.

Safety cues

The assessor may have served as an implicit safety cue. Certain social interactions are thought to provide safety cues, thus decreasing the threat value of the situation (Lohr, Olatunji & Sawchuk, 2007; Tang et al., 2007). A study investigating the influence of the presence of an observer and threatening information on pain reported during a cold pressor task found that, under the neutral information condition (i.e. when no threatening information was given to participants about the cold pressor task), there was no influence of the presence of an observer on reported pain. However, under a condition of threat (i.e. when participants were given threatening information about the cold pressor task), participants reported greater pain severity while facial expressions of pain were inhibited when no neutral observer was present than when a neutral observer was present during the procedure (Vlaeyen et al., 2009). This suggests that the observer may have acted as a safety cue in the presence of a threat manipulation. However, an alternative explanation for an inhibited facial expression is that the observer act as a threat cue, restricting communication of pain severity (Karos, 2018; Peeters & Vlaeyen, 2011). In the current study, participants reported pain severity verbally, but reported anxiety and threat of tissue damage on a computer, where the screen was not visible to the assessor. If the assessor acted as a threat cue in this current study, there would likely have been a dissociation between the verbal and computer-based manipulation checks i.e. there would have been decreased verbal pain ratings but increased computer-based ratings of anxiety and threat of tissue damage. Since this was not the case, and the assessor present when the participant received the threatening information, it seems more likely that the assessor acted as an implicit safety cue. The current study provided no data to which this possibility can be held up. Given that few participants reported not being anxious at all, it is more likely that safety cues than competing threat underlie the failure of the threat manipulation. However, this implicit safety cueing may have decreased the threat value of the sham skin examination and report, thus reducing the influence of threat on magnitude and surface area of secondary hyperalgesia at the threat site.

Trust, and requirements for participant safety

Our manipulation check results may reflect participants’ trust in the researchers, and the safety requirements for the procedure. Explicitly stating, in the study information, that the procedure is well-established and safe may have opposed the threat manipulation. This statement was a requirement of the Human Research Ethics Committee: “This procedure involves some pain; however, it is a well-established procedure and is known not to cause any skin damage”. Seven (of 26) participants cited trust in the researchers during the semi-structured interview. Specifically, one participant reported that they trusted that enough precautions had been taken to ensure the safety of the procedure. Another participant reported that they trusted the Human Research Ethics Committee would not approve an experiment that could cause damage to participants’ skin. Explicitly reassuring participants of the safely of the HFS procedure could have reduced the plausibility of the sham skin examination and report, thus reducing any influence of the manipulation on magnitude and surface area of secondary hyperalgesia.

Sampling bias

Our manipulation check results may reflect sampling bias. Our sample is unlikely to be representative of the general population. Low fear of pain and older age are associated with greater willingness to volunteer for a pain-related study (Karos, Alleva & Peters, 2018). If this finding extends to our context, low fear of pain in our sample may have opposed our manipulation. On the other hand, our manipulation was intended to be about tissue damage, not pain. The relevance of the findings of Karos et al. (2018) to our context is unclear: both studies included undergraduate students but from different countries, and under different compensation conditions. Karos et al. (2018) recruited students in Belgium, who must participate in research for course credit. We recruited any healthy control in South Africa, where research participation is not mandatory. To our knowledge, there are no published data on the characteristics of individuals who opt into or out of experimental pain research in South Africa. Such data would be useful to shed light on potential sampling bias in experimental pain studies and inform strategies to limit that bias.

Sample-specific habituation to threat

Our manipulation check results may reflect habitual exposure to threat in our sample. Many South Africans are regularly exposed to contextual threats (Hinsberger et al., 2016): one in three South Africans feels unsafe walking alone at night (Statistics South Africa, 2019), and continuous traumatic stress is common, given the frequency of domestic violence, family murders, gangsterism, and physical and sexual assault (Frenkel, Swartz & Bantjes, 2018; van der Merwe & Kassan-Newton, 2007). In the absence of informative data, we speculate that repetitive exposure to such contextual threats may contribute to pain-related neural processes, such as more efficient descending inhibition when exposed to threat. Further, repetitive exposure to threat has been positively associated with resilience (Scali et al., 2012). Therefore, high individual resilience may have opposed our threat manipulation strategy, particularly given the relatively safe laboratory environment. In fact, an exploratory comparison showed our participants’ CD-RISC scores (mean(range) 40.81 (32–48)) to be higher than normative data from an international sample of students and young adults (20.8–33.5) (Campbell-Sills & Stein, 2007; Hartley, 2012; Jones et al., 2017; Rahimi et al., 2014; Reyes et al., 2018; Shlomi, 2010). Further investigation of the relationship between trait resilience and resistance to experimental manipulations of threat would be useful.

Implausibility of (sham) skin examination and report

Finally, our manipulation check results may reflect the (im)plausibility of the (sham) skin examination and report. Participants may have considered it implausible that the skin on their one forearm was robust while the skin on their other forearm was fragile. However, this was not formally assessed. One participant reported that they found the use of the otoscope to examine the skin “rather odd” (although the assessor explained that the otoscope was used because of its light and magnification properties, allowing proper visualization of the skin). If the sham skin examination and report were not believable, it would have reduced the threat value associated with the HFS at the threat site and thus reduced the influence of the manipulation on magnitude and area of secondary hyperalgesia.

The need for effective threat manipulations for experimental pain research

There is a large gap in the literature relating to threat; although many researchers and clinicians invoke threat as an important concept in pain (Crombez et al., 1998; Karos et al., 2018; Reicherts et al., 2016; Tabor et al., 2015), threat has not been clearly defined and operationalized in the context of pain. Improved strategies are needed to define and measure threat associated with pain. Moreover, it is unclear whether different types of threat influence different physiological processes associated with pain. Implicit and explicit cues about a stimulus have been shown to change pain, and expected stimulus intensity affects pain (Arntz & Claassens, 2004; Moseley & Arntz, 2007). There are many candidate mechanisms by which threat may influence pain (e.g. decreased descending inhibitory control and increased ascending facilitation), but there are limited data testing these candidates. Therefore, to inform careful and effective targeting of therapeutic pain treatments, there is a need to clarify types of threat and the physiological and psychological mechanisms associated with different types of threat.

Optimizing the threat manipulation

Inducing a threat manipulation in a laboratory setting is known to be difficult; yet there are strategies to improve the effectiveness of threat manipulations. Threat manipulations are known to induce “weak…concerns about the pain stimulus” in experimental pain studies (Vlaeyen et al., 2009)—perhaps because participants know the pain will be short-lived and because ethical review provides implicit reassurance. Threat manipulations that give participants threatening information about the experimental procedures have been successful in previous studies (Jackson et al., 2005; Torta et al., 2019; Van Damme et al., 2008; Wiech et al., 2010). However, our early piloting of a strategy in which we provided participants with threatening information about the HFS procedure (rather than the integrity of the skin at the induction site) was ineffective in eliciting threat of tissue damage (see File S1: Piloting procedure).

To improve the effectiveness of the sham skin examination and report, we propose three modifications. First, studies could be structured for a between-group, rather than within-subject, comparison so that the threat value of anticipation of the HFS at the second site does not compete with the threat value of the sham skin examination. A comparison between sensory testing results before and after HFS would provide the outcome. Additionally, if participants thought it implausible to have “fragile” skin on the one forearm and “robust” skin on the other forearm, it may be more compelling if the (sham) skin examination and report were conducted on one arm only, with the other arm not being examined at all. Alternatively, a (sham) cream (e.g. Vaseline) could be applied to the skin on one forearm with information that this (sham) cream will make the skin more fragile/robust. Second, the social context could be adjusted in that the assessor is not in the room when the participant receives the threat manipulation (i.e. the results of the sham skin examination) so that the researcher does not act as a safety cue. Third, the statement that HFS is known not to cause any skin damage could be removed from the study information sheet, subject to agreement from the ethics committee.

Summed trauma scores and area of secondary hyperalgesia

Summed trauma scores were not correlated with surface area of secondary hyperalgesia in the current study. This conflicts with published pilot data in which summed trauma scores were positively associated with increased surface area but not increased magnitude of secondary hyperalgesia (You, Creech & Meagher, 2016). Importantly, the current study was not fully powered to detect this relationship. A possible reason for the conflicting results may be that our participants had lower summed trauma scores than those in the work by You, Creech & Meagher (2016).

We propose further experimental studies in the South African context (and other contexts with high rates of trauma) formally comparing (1) the magnitude of experimentally induced secondary hyperalgesia in participants with and without a history of trauma, (2) the effectiveness of different threat manipulations in participants with and without a history of trauma, and (3) the influence of a threat manipulation on the magnitude of experimentally induced secondary hyperalgesia in participants with and without a history of trauma.

Strengths

In the current study, we included manipulation checks assessing both implicit and explicit threat of tissue damage induced by our sham skin examination and report. The protocol was locked online and any deviations to the protocol have been declared here, thus supporting accountability and study replication (Lee et al., 2018). We conducted semi-structured interviews with participants and gained insight into the possibilities as to why our sham skin examination and report was unsuccessful. Additionally, this discussion provides a comprehensive overview of the challenges associated with conducting a threat manipulation for experimental pain research, which will be of benefit to researchers when designing a threat manipulation. This study also highlights the need and provides recommendations for future research investigating the association between threat and chronic pain among South Africans.

Limitations

An obvious limitation of this study is that the threat manipulation was ineffective. Therefore, whether threat of tissue damage is associated with greater magnitude and area of secondary hyperalgesia remains unanswered. Additionally, this study was not fully powered to detect the relationship between summed trauma scores and area of secondary hyperalgesia. Finally, inclusion of a psychophysiological outcome that could indicate implicit threat, such as heart rate, skin conductance response, or acoustic startle response, could have clarified the influence of the manipulation on implicit threat, and is suggested for future work.

Conclusion

The current study found that an adapted version of a previously successful threat manipulation (sham skin examination and report) was ineffective in eliciting a differential threat of tissue damage. Unsurprisingly, the primary analysis confirmed that neither magnitude nor area of secondary hyperalgesia was predicted by condition (i.e. which arm received the HFS under the supposedly threatening condition). We have extensively discussed opportunities to develop effective threat manipulations for experimental pain research, which we hope will be of benefit to the research community in taking this line of inquiry forward.

The current study also did not find a relationship between summed trauma scores and surface area of secondary hyperalgesia. This conflicts with published pilot data in which summed trauma scores were correlated with increased surface area but not increased magnitude of secondary hyperalgesia (You, Creech & Meagher, 2016). However, the current study was not fully powered to detect this relationship. Further research is required to clarify the potential relationship between trauma history and the magnitude and area of secondary hyperalgesia.

Supplemental Information

Supplemental Information 1 Piloting procedure.

Click here for additional data file.

Supplemental Information 2 Arrangement of electrodes.

Cathode is secured with double-sided tape to the participant’s anterior forearm and anode secured with Velcro to participant’s mid-upper arm. Photo credit: Gillian J Bedwell.

Click here for additional data file.

Supplemental Information 3 Sensation and Pain Rating Scale.

Click here for additional data file.

Supplemental Information 4 Eight radial lines approach.

(a) An image of the eight radial lines that originate at the centre of the site of the electrode. Each line is at a 45° angle to its neighbours, and each dots are 1 cm apart. (b) An example of a mapped area of secondary hyperalgesia. The green lines indicate the border of the area of secondary hyperalgesia.

Click here for additional data file.

Supplemental Information 5 R Markdown used for data analysis.

Click here for additional data file.

Supplemental Information 6 Script used by the assessor during the procedure.

Click here for additional data file.

Supplemental Information 7 Semi-structured interview.

Click here for additional data file.

Supplemental Information 8 Metadata.

Click here for additional data file.

Supplemental Information 9 Master data.

Click here for additional data file.

Supplemental Information 10 Demographic information.

Click here for additional data file.

Supplemental Information 11 Assessor blinding assessment.

Click here for additional data file.

Supplemental Information 12 CONSORT checklist.

Click here for additional data file.

Supplemental Information 13 CONSORT flow diagram.

Click here for additional data file.

The authors thank Mr Jeroen Clarysse for his technical assistance with automating this experiment and Dr Kessie Govender for making the cathodes.

Additional Information and Declarations

Competing Interests

Author Contributions

Human Ethics

Data Availability

1 Equivalent to USD6.18 at the time of the study.

Gillian J. Bedwell receives speakers’ fees for talks on pain and rehabilitation. Victoria J. Madden receives speakers’ fees for talks on pain and rehabilitation. Romy Parker receives speakers’ fees for talks on pain and rehabilitation, is a director of the not for profit organization Train Pain Academy, and serves as a councilor for the International Association for the Study of Pain. G. Lorimer Moseley has received support from: Reality Health, ConnectHealth UK, Seqirus, Kaiser Permanente, Workers’ Compensation Boards in Australia, Europe and North America, AIA Australia, the International Olympic Committee, Port Adelaide Football Club, Arsenal Football Club. Professional and scientific bodies have reimbursed him for travel costs related to presentation of research on pain at scientific conferences/symposia. He has received speaker fees for lectures on pain and rehabilitation. He receives book royalties from NOI group publications, Dancing Giraffe Press & OPTP for books on pain and rehabilitation. Caron Louw and Johan W. Vlaeyen have no conflicts of interest to declare.

G. Lorimer Moseley is an Academic Editor for PeerJ.

Gillian J. Bedwell performed the experiments, analyzed the data, prepared figures and/or tables, authored or reviewed drafts of the article, and approved the final draft.

Caron Louw performed the experiments, authored or reviewed drafts of the article, and approved the final draft.

Romy Parker conceived and designed the experiments, authored or reviewed drafts of the article, and approved the final draft.

Emanuel van den Broeke conceived and designed the experiments, authored or reviewed drafts of the article, and approved the final draft.

Johan W. Vlaeyen conceived and designed the experiments, authored or reviewed drafts of the article, and approved the final draft.

G. Lorimer Moseley conceived and designed the experiments, authored or reviewed drafts of the article, and approved the final draft.

Victoria J. Madden conceived and designed the experiments, performed the experiments, analyzed the data, prepared figures and/or tables, authored or reviewed drafts of the article, and approved the final draft.

The following information was supplied relating to ethical approvals (i.e., approving body and any reference numbers):

Faculty of Health Sciences Human Research Ethics Committee (REF 498/2018), University of Cape Town.

The following information was supplied regarding data availability:

The raw data and code are available in the Supplemental Files.

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
