# Peer review of "The influence of a manipulation of threat on experimentally-induced secondary hyperalgesia"

_PeerJ, doi:10.7717/peerj.13512_

## Round 0.1 · original submission · Minor Revisions

Dear Authors

The manuscript has been reviewed by two experts in the field and their comments are very positive with respect to the submitted manuscript, however, there are some methodological aspects that both reviewers suggest should be clarified and I agree with them.

My opinion of the manuscript is that it is a study carried out with scientific rigour, good methodological quality and absolute transparency. I congratulate the authors and recommend that they make the changes suggested by the reviewers so that their manuscript is finally accepted.

·

Basic reporting

In general, I consider your reporting style to be of a very high quality. I am in strong support of the open science approaches throughout the manuscript, in particular, the pre-registration of your analytical pipeline. I found the detail of your method to be excellent, and this has made the manuscript a much easier paper to review.

I have a few details that I would like you to review below:

Major points: 1) Within your abstract and introduction, you state that your approach will target "physiological changes within the dorsal horn", yet all of your outcome measures are focusing on subjective ratings of pain, anxiety or detection. I suggest that you are not, in fact, targeting the scope of your study towards spinal physiology, but are instead using SH as an experimental metric for examining the subjective impact of sensitisation of neurons within the dorsal horn. While perhaps a subtle distinction, I think it's important to separate your approach from spinal physiology.
2) Despite your transparency, I am a little unclear about your SH protocol. What were the instructions given to your participants to calibrate the points? Were they providing a point-rating based on detection, sensation change or pain? This has implications for the interpretation of the variable itself. Additionally, how did you calculate the area of an 8-sided polygon?

Minor points: 1) Odd formatting at line 117- use of italics is unusual, and I suspect a typo of actives<->activates
2) While you cite seminal work, I found that there were basic points (such as pathology not equalling symptomology, or clinical instances of pain-free behaviour) that were using very out-dated references. If possible, update your references to include more recent research, which is highly relevant to the arguments you are making

Experimental design

As stated, your discussion and methodological critiques were very well written and contained a number of well thought through ideas.

The only edit I'd recommend (or clarification I'd request) is that you proposed a theory of a failed manipulation due to "competition between threats", and that essentially, the threat was equally raised across both arms for the participants, rather than unilaterally. Have you tested the correlation between the ratings across arms for your sample? This seems like a good way to test whether it was the case, and based your scatterplots, appears likely.

Can you also clarify whether SH was collected by the same experimenter for every participant? High variability in SH measurements has been previously reported, with low inter-rater reliability. It appears that it was the case as your "assessor" completed all sensory tests. But I want to confirm this

Out of interest, I’d perhaps suggest that one concept that is missing which may tackle the issues you’ve encountered is to attribute the risk (within a between groups design) to the equipment, rather than the design itself. i.e. If the participants were to think that a piece of equipment was faulty, and had been performing irregularly, you may be able to impose risk which is outside of the umbrella of protection of the ethics and removes the trust element that exists between participant and experimenter. I don't require this to be added to manuscript, unless you decide so, but it could represent a solution to your issue.

Validity of the findings

I have no concerns about the validity of your findings, mainly due to the excellent description of methodologies and transparency of your design and approaches. In particular, I am pleased to see null findings being appropriately published.

Additional comments

I believe there are some naming issues concerning the supplimentaries. Line 550 refers to supplementary 5 for the results of your semi-structured interview, but this appears to be contained within supplementary 4. Likewise, on line 557 there is reference to supplementary 6, which was not included within my submission files. I suspect this will be a matter of formality, but I'd appreciate the opportunity to review these on re-submission

The reference section was irregularly formatted, different font type and size, and a few issues of punctuation and spacing

Again, out of curiosity, what was the rationale behind using an otoscope for a forearm inspection for threat? I did find this to be unusual from the off, and would have thought that most participants would be familiar with ear exams enough to suspect this was a sham inspection? You have addressed this limitation comprehensively, and I do not think this requires any more alteration to the manuscript.

·

Basic reporting

Typos etc.
• There are several double spaces that have crept in throughout the manuscript. I suggest using the replace function in Word.
• Line 112. The citation is in italics.
• Line 226. Should be “using a two millisecond pulse”.
• Line 743. Your references are font size 18. PeerJ’s guidelines say that text should be 12 points Times.
• Line 443. “larger surface area but magnitude” – the “but”.

Experimental design

Rigorous investigation performed to a high technical & ethical standard.
Some more detail is needed when describing the method (see 4. Additional comments).

Validity of the findings

All underlying data have been provided.
Conclusions are well stated, linked to original research question & limited to supporting results.

Additional comments

Bedwell and her colleagues examined the influence of a threat manipulation on induced secondary hyperalgesia from high-frequency electrical stimulation (HFS). Their experiment was preregistered and well designed. They aimed to induce threat by telling participants that one arm was particularly sensitive to HFS, and that the other one was not, and measure the effect on secondary hyperalgesia from HFS. However, it appears that the manipulation failed and/or the unpleasantness of HFS washed out any effect of the threat manipulation. It is a shame that the manipulation did not work, as the research question the authors set out to address is interesting and important. They had also gone to great efforts to ensure the that the study was rigorously designed (e.g. the blinding was done to a high standard).

In general, I think this is a well conducted study. The main issue is the threat manipulation, as the authors already point out. I believe there are a few points that need addressing/clarifying prior publication.


Major:
1. The discussion is lengthy (almost 3000 words). The authors raised many valid points and considered many different reasons why their manipulation may not have worked. I would suggest shortening the discussion and making their points more succinctly. Doing so will increase the readability of the manuscript, so that the interesting points that you raise are more likely to come across and not get missed out by your audience.
2. It would be helpful to include more details about the HFS stimulation. For instance, how does your procedure compare to previous studies? Did you measure the electrical detection threshold on one arm only? I am familiar with some of Emanuel van den Broeke’s previous work with HFS where they measure electrical detection thresholds for each arm, and show that there is some intra individual variability (e.g. Filbrich et al., 2020, Cortex). From reading your manuscript I am under the impression that thresholds were only measured for one arm. Is this correct? Either way, more detail is needed.
3. Related to my previous point, I think it would be helpful to include some of the information from Figure 4 in plain text in addition to its graphical presentation. It was not immediately clear to me how long after the HFS stimulation that the sensory testing etc. was conducted.
4. Is it necessary to abbreviate secondary hyperalgesia? The text would be easier to follow if you spell it out.
5. The effect size derived from your pilot data should be stated so that it is clear what magnitude of effect your study is powered to detect. Is it derived from the third stage of piloting (n = 6)?


Minor:
• It was not clear to me if there why the sensory testing was repeated so many times (e.g. 3 times at baseline, 6 times during follow-up). Please could you clarify the rational for doing so.
• Line 89. It would be useful to cite the three studies that you make reference to. Later you only go on to describing two of the three studies.
• Line 129. It would be helpful to outline how threat will be manipulated and what will be compared. It is difficult for the reader to contextualize the outcomes and hypothesis without this information. It should be made clear exactly what conditions etc. that will be compared.
• Line 244. How many pain ratings did participants give? Perhaps this information could also be depicted in Figure 4.
• Line 249. There is no clear rationale for why you are using pain ratings as your primary manipulation check. Please clarify. I suspect this is because you did not include a direct (quantitative) measure of perceived threat from the HEFs, and so had to rely on proxy measures.
• Line 258. How was anxiety measured?
• Line 727. More detail is needed about the procedure used to determine the primary and secondary outcomes (i.e. mechanical punctuate stimulation and surface area of secondary hyperalgesia).
• Line 325. Which arm was the individual threshold calibrated for?
• Line 343. Did any participants withdraw? It would be useful to report the number that withdrew, even if this is zero.
• Line 354. What was the motivation for measuring the surface area of secondary hyperalgesia three times?
• Line 379. What is the advantage of using a mean pain rating for the two pinpricks (i.e. 128 and 256 mN) rather than having the raw ratings for each and adding Force (128 mN, 256 mN) as a fixed factor? You approach might be equivalent, or even superior. Either way it would be useful to add some detail so that the reader understands the reasoning behind this decision.
• Line 392. PeerJ’s guidelines for citing unpublished work states: “Cite work unpublished, in preparation or under review as 'unpublished data'. Supply the author's first initial and surname, and the year of the data collection, in the text citation and do not include the citation in the reference section. Example: (A Castillo, 2000, unpublished data).” Please correct your citation.
• Line 395. Was there a correlation between individually determined thresholds and magnitude of secondary hyperalgesia in your unpublished data?
• Line 435. How were p-values calculated from your linear mixed model? E.g. did you use the Satterthwaite degrees of freedom (see Luke, 2017, Behavior Research Methods)?
• Line 456. The information about what the box plots show etc. should be included in the legend of each figure containing a boxplot so that the figures can be interpreted on their own.
• Line 480/Figure 5. Is it important to show the difference between HFS trains? This figure might be easier to read if you simply have condition on the x-axis (e.g. with a box plot). This point also applies to Figure 8.
• Line 496. More detail in the figure caption would be helpful. For instance, what does a positive value show on the y-axis show?
• Line 625. A citation might be needed here. Has it been demonstrated that knowledge of a monetary reward activates descending modulatory mechanisms?

---

## Round 0.2 · Minor Revisions

Dear Authors

I congratulate you for the great effort to revise the manuscript and respond to all the recommendations suggested by the reviewers. The manuscript has been substantially improved and I would like to announce that it is accepted for publication in this journal.

One of the reviewers has raised some minor issues that the authors feel could further improve the manuscript and its reproducibility if considered by the authors.

If you agree to make these changes please send us a new version of the manuscript.

Best regards and congratulations

·

Basic reporting

Thank you for addressing my previous comments. I only have very minor comments that address some of the revisions made to the manuscript.

Figures 6 & 7 have a legend for the Alpha. I imagine this in unintentional, or possibly to distinguish the outliers from the box plots. If it is the latter, you can simply suppress the outliers of the box plot in ggplot with this line of code: "geom_boxplot(outlier.shape = NA)+". If, however, the different alpha levels are intentional, then their significance should be described in the caption.

Typos:
Line 253. You should define SPARS the first time the acronym is used (i.e. line 253 rather than 273).
Line 264. It is not entirely clear what "the procedure" refers to. Presumably this refers to your experimental protocol, as outlined in figure 2.
Line 482. There is a space missing: “the‘lmer’”.


*line numbers refer to the revised manuscript with tracked changes.

Experimental design

The authors provided an in depth response to my question about how the sample size was calculated (i.e. R2.13). I would like to see more of this justification in the manuscript. For instance, there is no mention of scale factor in the manuscript. As a consequence, it is not clear from the manuscript how the pilot data (n = 6/third phase) was used to inform the sample size estimate. This issue could be addressed by adding more detail to the main manuscript, or to include a full breakdown of your GLIMMSE calculation as a supplementary file.

I appreciate that this is somewhat pedantic, but I consider it important detail to ensure that your study can be replicated.

Validity of the findings

The preregistration is still under embargo. Otherwise I have no comments to add.

Additional comments

I congratulate the authors on their effort to revise the manuscript. I believe it has improved substantially and is appropriate for publication. I have raised a few minor issues that the authors might want to address to further improve the manuscript and its reproducibility.

---

## Round 0.3 · accepted · Accept

Dear Authors

I am pleased to inform you that your manuscript is finally accepted for publication. Your study meets the methodological and scientific requirements for publication in this journal, and I congratulate you for all the effort made during the review period.

We hope that you will continue to consider this journal for future manuscript submissions.

Best regards